# The Programmable Design of Large-Area Piezoresistive Textile Sensors Using Manufacturing by Jacquard Processing

**DOI:** 10.3390/polym15010078

**Published:** 2022-12-25

**Authors:** SangUn Kim, TranThuyNga Truong, JunHyuk Jang, Jooyong Kim

**Affiliations:** Department of Smart Wearables Engineering, Soongsil University, Seoul 06978, Republic of Korea

**Keywords:** textile sensors, electro-textile, piezoresistive sensor, Jacquard processing, wearable electronics

## Abstract

Among wearable e-textiles, conductive textile yarns are of particular interest because they can be used as flexible and wearable sensors without affecting the usual properties and comfort of the textiles. Firstly, this study proposed three types of piezoresistive textile sensors, namely, single-layer, double-layer, and quadruple-layer, to be made by the Jacquard processing method. This method enables the programmable design of the sensor’s structure and customizes the sensor’s sensitivity to work more efficiently in personalized applications. Secondly, the sensor range and coefficient of determination showed that the sensor is reliable and suitable for many applications. The dimensions of the proposed sensors are 20 × 20 cm, and the thicknesses are under 0.52 mm. The entire area of the sensor is a pressure-sensitive spot. Thirdly, the effect of layer density on the performance of the sensors showed that the single-layer pressure sensor has a thinner thickness and faster response time than the multilayer pressure sensor. Moreover, the sensors have a quick response time (<50 ms) and small hysteresis. Finally, the hysteresis will increase according to the number of conductive layers. Many tests were carried out, which can provide an excellent knowledge database in the context of large-area piezoresistive textile sensors using manufacturing by Jacquard processing. The effects of the percolation of CNTs, thickness, and sheet resistance on the performance of sensors were investigated. The structural and surface morphology of coating samples and SWCNTs were evaluated by using a scanning electron microscope. The structure of the proposed sensor is expected to be an essential step toward realizing wearable signal sensing for next-generation personalized applications.

## 1. Introduction

In recent years, wearable sensors called wearable electronic textiles have widely been investigated for personalized applications such as monitoring biomedical and physiological signals [1,2,3]. They are flexible, comfortable, lightweight, and can maintain the desired electrical property. Additionally, advances in flexible force-sensitive materials are enabling the fast development of smart fields to convert mechanical stimuli to electrical signals for promising applications in electronic skin (e-skin), wearable electronics, health monitoring (human heart rate, weight, movement), etc. [4,5]. There are various sensors, such as pressure, strain, muscle activity, electrocardiogram (ECG), etc. Smart textile sensors with high sensitivity and stretchability being driven at low power consumption are desirable. Due to highly sensitive responses, low energy consumption, and simple fabrication processes, piezoresistive pressure sensors attract much attention and exhibit promising potential [6,7,8,9,10]. Significantly, one of the exciting characteristics of textile pressure sensors is that sensor yarns can be implemented in soft electronics, including perceptive devices such as biological monitoring that are able to be fitted to curved surfaces. For example, a study applied the textile pressure sensor to shelves that were able to weigh products and determine whether the product was on the shelf, and then record the inventory history [11]. Another study presented a smart tablecloth to monitor dietary behavior using a textile pressure-sensor matrix facilitated by parallel stripes of metallically conductive threads as the key sensing element [12]. In that study, they investigated the distribution force on the surface underneath the plate. Finally, a smart tray connected with specific containers can detect the spots of food, categories, and numbers of meals for diet. These studies show that the sensing data of pressures can be converted into valuable sources for monitoring and detecting many kinds of information.

In many studies, a conductive thread is used in the embroidery process on woven fabric, or conductive solutions are applied to the surface to act as mechanical stimuli surfaces called electrodes [13,14,15]. These studies use Jacquard processing, one of the weaving methods that can be programmable and effective with a short process. Electrodes were designed and attached to fabric using this processing, enabling various and complex fabric structures and designs to thereby improve the sensitivity of biosignals [16,17]. From a manufacturing point of view, compared to the embroidery method, the advantages of the Jacquard method are its being fast, a flexible process, programmable, short time, low-cost, and making it easy to weave large-area textiles in the weaving process.

In previous studies [18,19], conductive silver thread and silver ink have been used as electrically conductive materials for e-textile applications due to their advantages, including excellent electrical conductivity, good mechanical properties, and high specific surface area. Herein, we demonstrate a pressure piezoresistive sensor (PPS) fabricated by the Jacquard weave structure. Three types of piezoresistive structures were designed using the Jacquard method that can program the characteristics of weave textile pressure sensors by changing different thicknesses, evidently allowing it to achieve multifunctional force sensing. The piezoresistive characteristics that followed the resistance equation of the pressure sensors were evaluated to determine whether they were suitable to detect loads for applications in many fields. The proposed sensor yarn has a distinctive characteristic of sensitivity to compression, but it is highly insensitive to free bending without tension. Moreover, the usefulness of these double-weaved structures was determined when applying a weight up to 1 kg with an error rate of 5% or less, and sensor durability could be verified by measuring a change of initial resistance in a repeated experiment of 1000 cycles against a high load of 98 N. This study includes the piezoresistive textile sensor’s programmable and effective manufacturing method, and the double-weaved structure is evaluated for its suitability to be used in various smart textile fields.

## 2. Experiments

### 2.1. Manufacturing of Jacquard Piezoresistive Textile Pressure Sensor

In this study, for the conduction of the Jacquard textile pressure sensor, 140D Nylon 6 yarn was used as an electrode yarn core and coated with silver ions. After that, the textile electrodes (yellow color can be seen in Figure 1) were woven by one of four Jacquard looms (RP9500 QRP model from Itema and Bonas 800, Colzate, Italy) were used for the Jacquard loom and controller. As shown in Figure 1, the pressure sensor section is made with three of the four looms and is woven into three preset structures in the order of single layer (a), double layer (b), and quadruple layer (c). The structure of textile electrodes was designed by applying 3D structure models through WiseTex, a software that can model the fabric structure [20].

As shown in Figure 2b–d, yarn fibers were interlaced into three types with three different thicknesses from low to high, respectively, 0.16 mm, 0.36 mm, and 0.52 mm. The properties of each type will be examined in more detail in the next section.

The gray yarns are the non-conductive or regular yarn, and the yellow threads are weft silver conductive yarn in the 3D structure models. The process of sensing layers impregnated with carbon nanotubes (CNTs) ink dispersed in methyl ethyl ketone (MEK) depends on agglomerating by the Van Der Waals force of the surface. The CNTs ink was stirred for at least 1 h with a spin speed of 700 rpm; then, the compound liquid was put in an ultrasonic machine to avoid incorporating air bubbles to ensure uniform distribution on the fabric’s surface. The solution was poured on a plate, and the samples were dip-coated. The conductive particles were evenly spread through the padding machine. After this process, the material impregnated in CNTs ink was dried for around 15 min at 100 degrees Celsius to establish CNTs particles adhering to the fabric (shown in Figure 3).

A scanning electron microscope (Gemini SEM 300 from ZEISS, Oberkochen, Germany) was used to evaluate the structural and surface morphology of coating samples and SWCNTs. As can be seen in Figure 4, m-SWCNTs have a diameter ranging from 1 to 1.5 nm, as obtained from the previous study [21]. We used single-walled carbon nanotubes (m-SWCNTs) due to their carboxyl (-COOH) functional groups improving the bonding between SWCNT and fiber yarn. SEM microphotographs from 2.5 k to 70 k magnification of coating samples are shown in Figure 4. As can be seen, the CNTs were well-dispersed around the surface of the filament yarn.

### 2.2. Electrical Properties

To analyze the characteristics of pressure sensors, we used an LCR E4980AL meter from Keysight connected to a universal testing machine (UTM) from Dacell Co. Ltd., Seoul, Republic of Korea (shown in Figure 5a). The sample under test (SUT) was put into the space for the compressive specimen. The electrical wires of the LCR meter were connected directly to the electrodes shown in Figure 5a. In order to achieve the proper measurements, we kept the same settings in all testing. The resistance and pressure values were measured under the loading force and transferred to the computer. At least ten samples for each structure introduced in Section 2.1 were tested and we recorded the averages.

The working principle shown in Figure 4b,c is for a single-layer pressure sensor, which is based on the percolation theory using CNT particles as filler load to enhance the conductivity of ordinary yarn. In this theory, two main issues should be considered. The first is interactions between CNT particles, known as the percolation path (shown in Figure 6).

Moreover, as we can see in Figure 5c, $d_{eo}$ is a gap between two CNT particles. $d_{0}$ is the gap between CNT particle and the electrode. Under pressure, these separations narrow, resulting in increased density of distribution and dispersion of CNT particles inside the yarn filament, which leads to the second issue, concentration. Accordingly, the conductivity will increase when the density of CNT particles increases. As far as we know, conductivity is the inverse of resistivity. Therefore, based on sensing the change in electrical resistance, when these distances become closer, the electrical resistivity ρ (rho) (also called specific electrical resistance) of material, measured in ohm-meter (Ω.m), decreases, leading to a reduction in the resistance, computed as (1) below:(1)$$R=\rho \frac{l}{A}$$
where *l* is the fiber length, and *A* is the cross-section area.

Note that, under pressure, the cross-section area also decreases, resulting in increased resistance. However, in this case, the fibers are woven in thin layers forming a conductive sheet. Therefore, the reduction in parameter *A* is negligible compared to electrical resistivity. Finally, the total resistance of textile pressure sensors decreases with the pressure. As a result, the arrangement of conductive particles or substances in the matrix changes, and the overall resistance decreases, so the pressure can be measured [13,14,15,22].

## 3. Results

In order to study the effect of the number of layers on the performance of piezoresistive textile sensors, we tested samples under loads from 0 to 2 kg. The sensor size was 20 × 20 cm, and the length between the two electrodes was 10 cm. The proposed sensors’ resistance changes were measured and recorded via an LCR meter connected to the UTM, as shown in Figure 7. In addition, we used Digilent Analog Discovery 2 to recheck data. In this experiment, the goal was to demonstrate the effect of the number of conductive layers on the performance of PPT. Moreover, by using Diligent, our aim was to show that the sensitivity of the sensor can be changed simply by varying the number of conductive layers stacked to suit the applications that will be shown in detail in the following study. Finally, the linear correlations between the resistance and the compression force applied to the sensors can be observed in Figure 8. As expected, results demonstrated an electrical resistance change from 2400 Ω at the unload pressure to 950 Ω at the load pressure.

The sensitivity (*S*) of a piezoresistive sensor is an essential parameter for the assessment of device performance and is characterized using the following equation:(2)$$S=\frac{R-R_{0}}{R_{0}}\times 100\%$$
where Δ*R* is the resistance change before and after the load is applied, and $R\;\mathrm{and}\;R_{0}$ are values of the sensor under load and without load, respectively.

As we can see, the changes in resistance for the single-layer sensor increased from 0 to around 350 ohms, larger than for the double and quadruple ones, which means that the single-layer sensor had higher sensitivity. More specifically, when we increase the number of conductive layers, the sensitivity also changes, but not linearly. This may be because the percolation or permeability inside the dual layer is higher than the single layer but lower than the fourth layer. The sensitivity of single, double, and quadruple layers calculated by (2) were 14.58%, 3.57%, and 5%, respectively.

In Figure 8, we used a linear function to describe the relationship between weight and resistance variables. The linear fit showed that the structure of the single-layer sensor had a good fit from 0 to 200 g, while double and quadruple ones had a more extensive range of up to 1000 g. Moreover, the linear fit plot also shows the establishment of a tunneling mechanism inside the fabric, which improves electrical conductivity. However, the structure of the single-layer sensor exhibited a more linear resistance-to-force correlation and higher sensitivity. In a comparison overview, our sensors were thinner than those of these other studies [23,24,25,26,27].

The error rates and graphs were obtained with test data for 0.2 kg in the single-layer sensor and 1 kg in the double-layer sensor, as summarized in Table 1. In all structures, the coefficients of determination showed a high value of 0.97 or more, but the test data error rate was 23.98% for the single-weave design and less than 5% for the double-weave structure.

After this, to investigate the durability of the sensors, more than 1000 cycles of loading cyclic tests were carried out. Figure 9 describes the resistance change of the sensors after 1000 loading cycles. As can be seen, the resistance increased but stabilized after some cyclic loading/unloading periods. Response time is an important parameter that defines the time lag between an electronic input and the output signal. Figure 9b shows that increasing the number of conductive layers from single to double improved the stability of the sensor, but that it also reduced the response time of the sensor (shown in Figure 9c). This is due to the overlap of the sensor layers under pressure and uneven rebound when the pressure force is reduced. In addition, the viscoelastic nature of wool yarns and the connectivity between the CNT coatings under pressure also cause the response delay in the sensor. Therefore, an increasing number of conductive layers, in this case, also increases the hysteresis of the sensor.

Moreover, the test data error rate for the single-weave design was higher due to the highest resistance of the single layer; its fluctuation under pressure will also be more significant than that of the other two types. The immediate connection time of conductive particles ensures the rapid electrical property of these sensors in particular applications. In addition, the sensors had a quick response time (<50 ms) at 200 g and 600 g with small hysteresis (shown in Figure 10a).

Moreover, a washing test (around 50 times) was carried out using a mini washing machine (LG-W0082) from Daewoong Co., Seoul, Republic of Korea. In the washing test, each washing time had a duration of 5 min, a squeezing time of 2 min, and a drying time of 7 min at 100 °C. The sensors demonstrated that the resistance increased slightly (less than 10%) after 50 washing times (Figure 10b). This is significant for wearable applications [28].

## 4. Conclusions

In this study, we made three types of sensor structures with a large area using the Jacquard process rather than the embroidery process used in the popular manufacturing method for textile sensors. Textile sensors with piezoresistive properties were made through the dip-coating process with SWCNTs known as conductive particles. Calibration graphs with high coefficients of determination between the resistance change value and the weight change were made to evaluate the performance of the actual sensor. Although the sensitivity of the proposed sensors was not high, we will present suitable applications for them in further studies owing to the relationship between the sensitivity and the number of conductive layers of the sensor. Finally, we found that the double-weaved structure sensors were suitable for manufacturing large-area pressure sensors. In the future, if studies on such a textile sensor manufacturing method proceed, it is expected that actual commercialization and related industries will be developed.

## Figures and Tables

**Figure 1 polymers-15-00078-f001:**
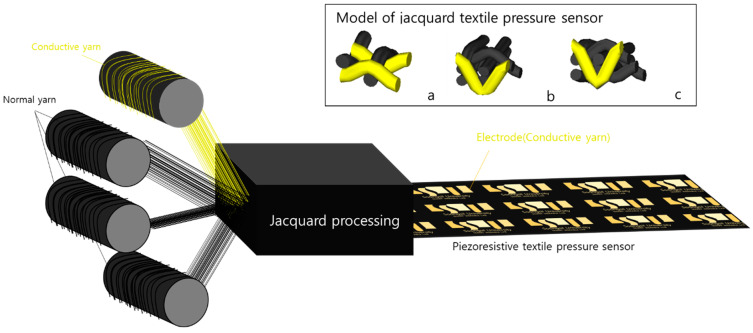
Manufacturing of 3 structure types of piezoresistive textile pressure sensors using Jacquard processing: (**a**) single layer, (**b**) double layer, and (**c**) quadruple layer.

**Figure 2 polymers-15-00078-f002:**
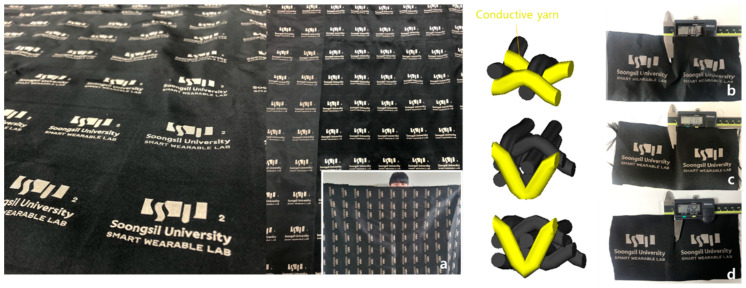
(**a**) Large-area piezoresistive textile pressure sensor, (**b**) single layer with thickness 0.16 mm, (**c**) double layer with thickness 0.36 mm, (**d**) quadruple layer with thickness 0.52 mm.

**Figure 3 polymers-15-00078-f003:**
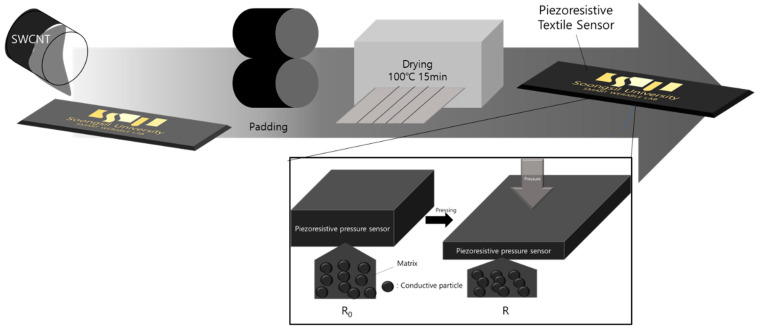
The fabrication process for dip coating of pressure-sensing layers.

**Figure 4 polymers-15-00078-f004:**
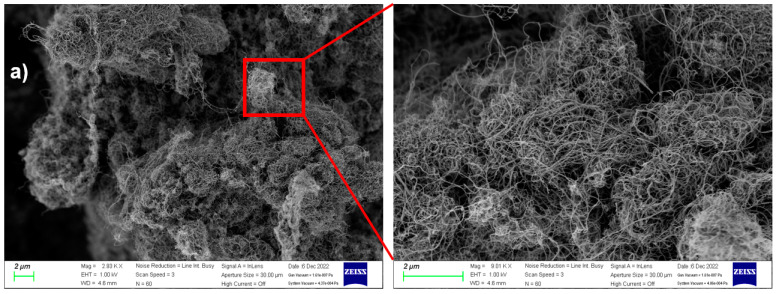
The SEM images of (**a**) SWCNT, (**b**) non-conductive fabric, and (**c**) conductive yarn.

**Figure 5 polymers-15-00078-f005:**
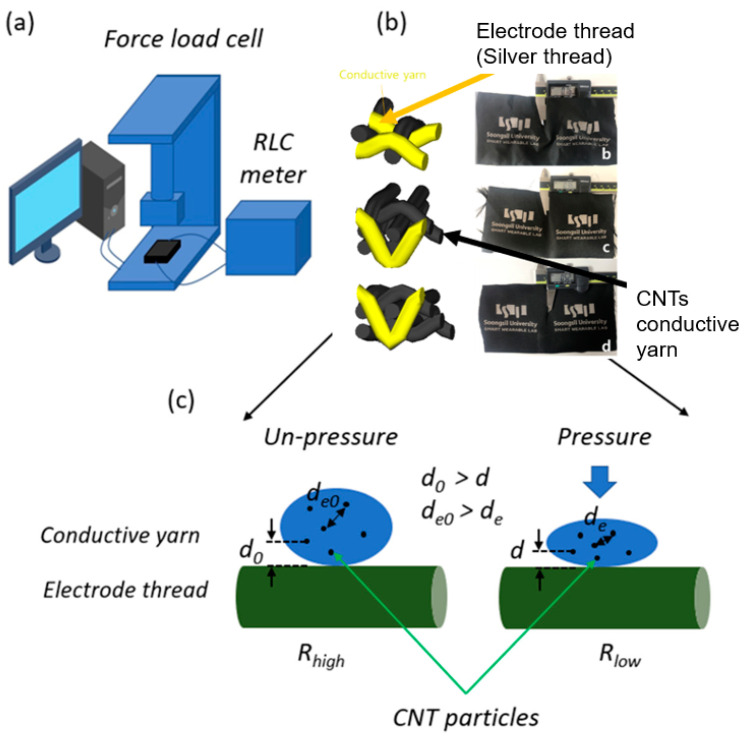
(**a**) Testing machine system, (**b**) piezoresistive textile sensors, (**c**) electrical properties.

**Figure 6 polymers-15-00078-f006:**
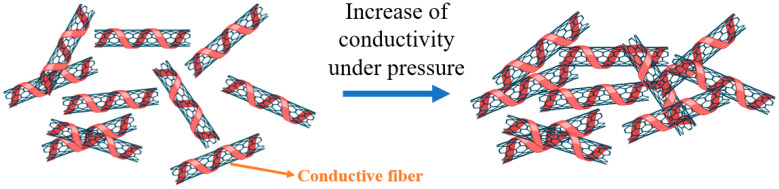
Schematic illustration of the conductive mechanism.

**Figure 7 polymers-15-00078-f007:**
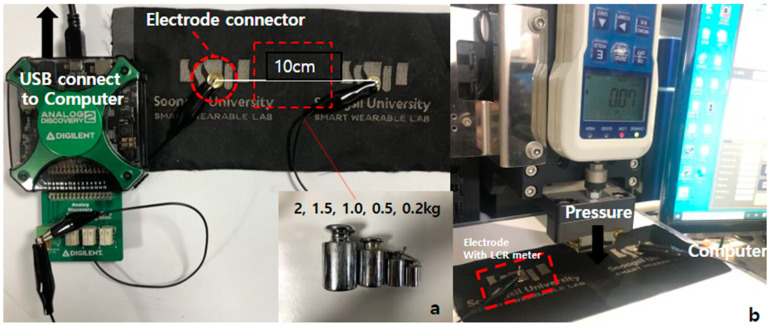
Measurement setup for a proposed pressure sensor, (**a**) loads, (**b**) UTM.

**Figure 8 polymers-15-00078-f008:**
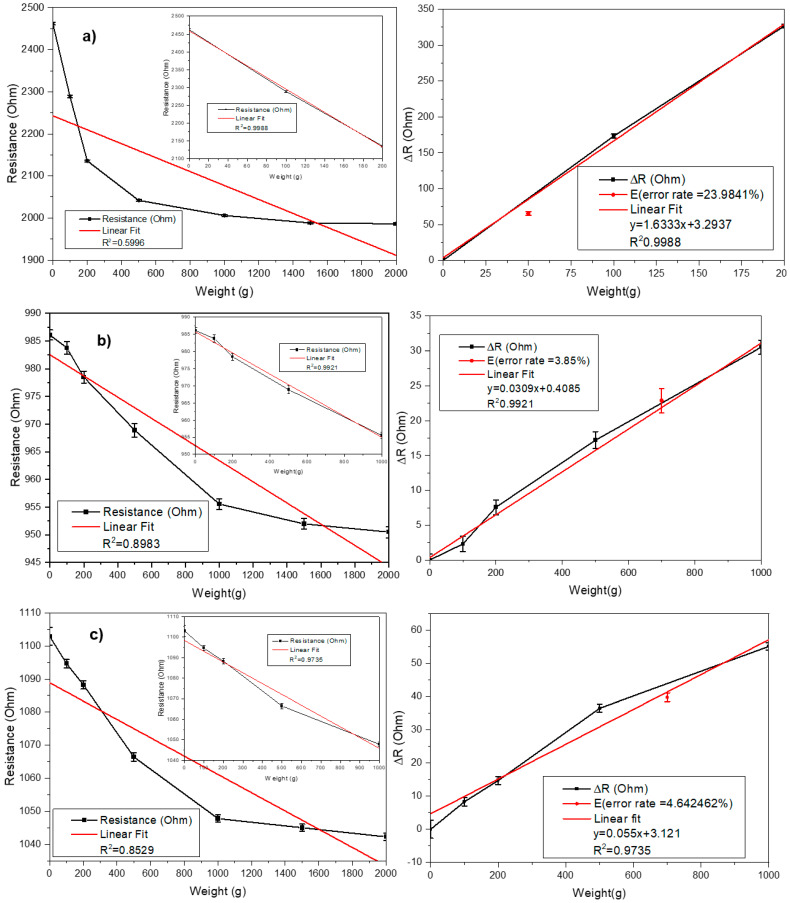
The compression test results for (**a**) single−layer, (**b**) double−layer, and (**c**) quadruple−layer sensors.

**Figure 9 polymers-15-00078-f009:**
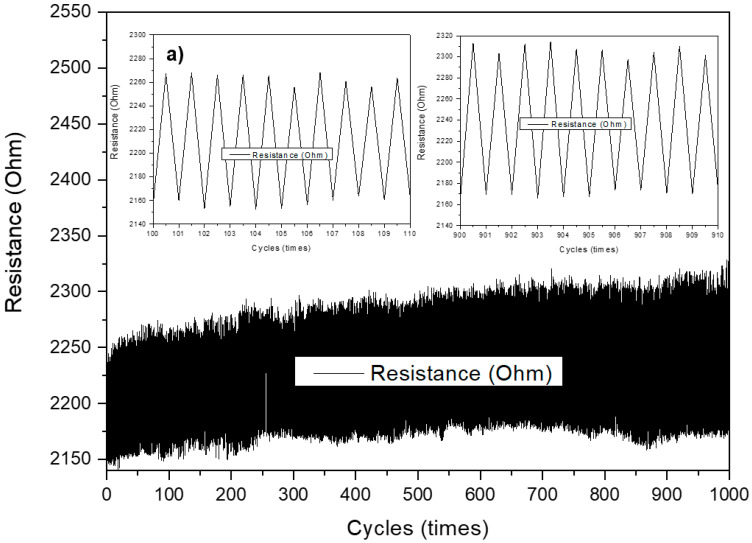
1000-times cyclic test results for (**a**) single-layer, (**b**) double-layer, and (**c**) quadruple-layer sensors.

**Figure 10 polymers-15-00078-f010:**
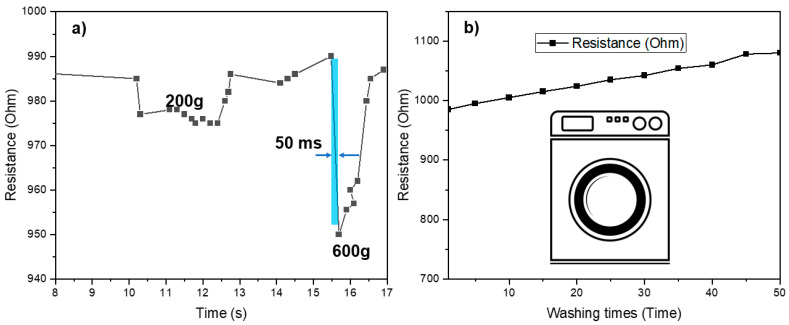
(**a**) Response time. (**b**) The resistance value of conductive nonwoven after 50 washing times.

**Table 1 polymers-15-00078-t001:** Piezoresistive textile pressure sensor calibration.

Structures	Equations	$\mathrm{Coefficient}\;(R^{2})$ (%)	Error Rates (%)
Single-layer	1.633x + 3.2937	0.9988	23.98
Double-layer	0.0309x + 0.4085	0.9921	3.85
Quadruple-layer	0.055x + 3.121	0.9735	4.64

## Data Availability

Not applicable.

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
