# Peer review of "The Programmable Design of Large-Area Piezoresistive Textile Sensors Using Manufacturing by Jacquard Processing"

_polymers, 2022, doi:10.3390/polym15010078_

Round 1

Reviewer 1 Report

This study suggests the three types of piezoresistive textile sensors, named single-layer, double-layer, and quadruple-layer, by the jacquard processing method to enable the programmable design of the sensor’s structure and customizes the sensor’s sensitivity to work more efficiently in personalized applications. A few comments appended below should be addressed prior to acceptance for publication in the Polymers Journal as follow:

1. In the abstract: the authors should present the significant findings.

2. Introduction: the novelty of this study should be elucidated.

3. Results: please improve the figures.

4. Conclusion: it is too long, please summarize it.

Author Response

Dear Reviewers and Editor,

We would like to thank the reviewers for the valuable comments, which are very helpful for revising and improving our manuscript. We appreciate the valuable feedback and have revised the manuscripts according to the comments provided by the reviewers as an attachment.

Reviewer 2 Report

The authors have presented large area piezoresistive textile sensors manufactured through Jacquard processing. Three types of sensors with different number of layers are fabricated and electrical properties are evaluated. The article is a proof-of-concept work with brief discussion of the electrical results. More experiments are needed to make the work comprehensive and interesting to the readers. This work could be re-considered for publication after major revisions.

Major comments:

1. The authors state in the abstract that the sensors have a quick response time (<50 ms) but this was not clearly supported with data plots. The authors should define what they consider as response time and highlight that in the current plots or make new plots.

2. The authors should present the power consumption of their sensors.

3. A common testing procedure for evaluating the durability and stability of textile sensors is a washing test. Please perform this test and present the results.

Reference: https://www.mdpi.com/2073-4360/14/8/1545

4. The authors state in the introduction “distinctive characteristic of sensitivity to compressive and tensile strains but is highly insensitive to free bending without tension”. Only compression test results are presented for sensitivity of the textiles, tension and bending tests should be performed and presented as well.

5. The sensitivity of the different layered sensors doesn’t follow a trend. Have the authors measured the resistance in a four-probe setup to eliminate the resistance of the wires? In high resistance measurements two-probe setups can cause incorrect resistance readings.

6. Why was the test data error rate for single-weave design higher?

7. The change in resistance for 1000 g load should be presented for single layer sensors as well.

Minor comments:

1. On line 86, the authors describe gray yarns as non-conductive and the orange threads are conductive however, the visible colors are gray and yellow. Please correct this.

2. On line 104, please refer the correct Figure number.

3. Please point to the correct threads for conductive and non-conductive yarns in Figure 5b, it is confusing right now.

4. Correct the subscripts in line 132 and 133.

5. Line 204, the sentence “The immediate connection time….” is not clear. What is particle application?

Author Response

(The authors gave the same response as above.)

Round 2

Reviewer 2 Report

The authors have adequately answered the queries and the new experiments have improved the manuscript. I recommend for publication.